# Blood cell image segmentation and classification: a systematic review

Muhammad Shahzad[1,*], Farman Ali[2,*], Syed Hamad Shirazi[1], Assad Rasheed[1], Awais Ahmad[3], Babar Shah[4] and Daehan Kwak[5]

[1] Department of Computer Science and Information Technology, Hazara University, Mansehra, Pakistan
[2] Department of Computer Science and Engineering, School of Convergence, Sungkyunkwan University, Seoul, South Korea
[3] Centre for Excellence in Information Technology, Institute of Management Sciences, Peshawar, Pakistan
[4] College of Technological Innovation, Zayed University, Dubai, United Arab Emirates
[5] Department of Computer Science and Technology, Kean University, Union, NJ, United States
* These authors contributed equally to this work.



Corresponding authors
Syed Hamad Shirazi,
syedhamad@hu.edu.pk
Daehan Kwak, dkwak@kean.edu

## ABSTRACT

**Background:** Blood diseases such as leukemia, anemia, lymphoma, and thalassemia are hematological disorders that relate to abnormalities in the morphology and concentration of blood elements, specifically white blood cells (WBC) and red blood cells (RBC). Accurate and efficient diagnosis of these conditions significantly depends on the expertise of hematologists and pathologists. To assist the pathologist in the diagnostic process, there has been growing interest in utilizing computer-aided diagnostic (CAD) techniques, particularly those using medical image processing and machine learning algorithms. Previous surveys in this domain have been narrowly focused, often only addressing specific areas like segmentation or classification but lacking a holistic view like segmentation, classification, feature extraction, dataset utilization, evaluation matrices, *etc.*

**Methodology:** This survey aims to provide a comprehensive and systematic review of existing literature and research work in the field of blood image analysis using deep learning techniques. It particularly focuses on medical image processing techniques and deep learning algorithms that excel in the morphological characterization of WBCs and RBCs. The review is structured to cover four main areas: segmentation techniques, classification methodologies, descriptive feature selection, evaluation parameters, and dataset selection for the analysis of WBCs and RBCs.

**Results:** Our analysis reveals several interesting trends and preferences among researchers. Regarding dataset selection, approximately 50% of research related to WBC segmentation and 60% for RBC segmentation opted for manually obtaining images rather than using a predefined dataset. When it comes to classification, 45% of the previous work on WBCs chose the ALL-IDB dataset, while a significant 73% of researchers focused on RBC classification decided to manually obtain images from medical institutions instead of utilizing predefined datasets. In terms of feature selection for classification, morphological features were the most popular, being chosen in 55% and 80% of studies related to WBC and RBC classification, respectively.

**Conclusion:** The diagnostic accuracy for blood-related diseases like leukemia, anemia, lymphoma, and thalassemia can be significantly enhanced through the effective use of CAD techniques, which have evolved considerably in recent years. This survey provides a broad and in-depth review of the techniques being employed,

from image segmentation to classification, feature selection, utilization of evaluation matrices, and dataset selection. The inconsistency in dataset selection suggests a need for standardized, high-quality datasets to strengthen the diagnostic capabilities of these techniques further. Additionally, the popularity of morphological features indicates that future research could further explore and innovate in this direction.

# INTRODUCTION

The microscopic examination and morphological evaluation of a peripheral blood smear are valuable diagnostic techniques for identifying blood-related disorders, including leukemia (*Mishra, Majhi & Sa, 2019*; *Shahzad et al., 2022*), anemia, thalassemia, *etc.* Most of them become fatal if not treated in time (*Mishra, Majhi & Sa, 2019*). So far, the blood syndrome diagnosis greatly relies on automatic analyzers, which have some limitations for abnormal or neoplastic cell detection (*Puigví & Alférez, 2018*). It is essential to develop computer-aided diagnostic (CAD) techniques that support hematologists to enable an accurate and efficient morphological analysis of blood elements. The primary function of a CAD system is to extract the underlying characteristics of blood elements that lie beyond the range of human visual perception. These features help to classify the blood elements, *i.e.*, white blood cells (WBC), red blood cells (RBC), and platelets, as healthy or affected. Although the diagnosis of blood-related diseases assisted greatly with cytogenetic, molecular, and immunological tests, physical and quantitative descriptors and features like morphology, texture, and color play a crucial role in diagnosing several blood-related diseases (*Puigví & Alférez, 2018*). Traditionally, peripheral blood smear analysis on slides is subjected to manual human inspection. This inspection process needs expert and well-trained personnel with an enormous amount of time to diagnose the disease. The manual blood morphology analysis is very tedious, long, subjected to error, and prone to hematologist expertise. Manual analysis requires tremendous proficiency, time, resources, and concentration. Lack of any can cause severe damage in the diagnostic analysis. This makes an automated blood cell image analysis essential for correctly and efficiently identifying blood-related abnormalities. Computer-aided blood cells morphological analysis, like segmentation and classification, are the central elements of machine-based screening of several blood diseases. Here, a question arises: *Is it conceivable to propose or develop an artificial intelligence (AI) system based on image processing techniques that can capture the morphological, texture, or color features of blood ingredients to predict blood cell disorder?* This question is still an open challenge for researchers to develop machine learning (ML) or deep learning (DL) based techniques for predicting abnormalities in blood elements. Several ML-based methods have been developed so far to beat this challenge. These ML-based models' main constituents are segmentation, feature extraction, and classification. Segmentation is a process in which an image is partitioned into a set of objects based on the region of interest (ROI). During feature extraction, a

group of measurable descriptors is identified for each ROI object (*Rodellar et al., 2018*), while classification identifies each ROI with a specific label.

## Motivation for study

Several survey works have been carried out in this domain (*Aldrin Karunaharan & Prakash, 2018*; *Bagasjvara et al., 2017*; *Das et al., 2022*; *Reyes, Rozo & Morales, 2015*; *Rodellar et al., 2018*; *Shafique & Tehsin, 2018b*) but most of them target few areas, *i.e.*, in *Rodellar et al. (2018)*, authors are mainly concerned with the morphological analysis of blood cells. This study focuses only on the WBCs-related disease analysis (malignant lymphoid and blast cells). However, there is no comparative analysis among techniques and algorithms used for blood cell segmentation and classification. While (*Reyes, Rozo & Morales, 2015*) carried out a review that was only concerned with leukocyte image segmentation and advantages and flows. They also do not focus on the classification and RBC segmentation approaches. Another study (*Puigví & Alférez, 2018*) was conducted to analyze those techniques related to the optimization of blood morphology for image analysis. They focused only on the 03 classes of descriptive features, *i.e.* geometrical, color, and texture features. Surveys on WBC segmentation have been carried out by *Patel & Prajapati (2018)* but only concerned with segmentation techniques. In *Abas, Abdulazeez & Zeebaree (2022)*, *Bagasjvara et al. (2017)*, *Das et al. (2022)*, *Shafique & Tehsin (2018b)*, the author works on segmentation and classification techniques regarding acute lymphoblastic leukemia (ALL) cells. They perform analysis only on ALL-IDB cells image analysis. Each research work mentioned above is confined to only a specific blood image analysis domain area. None of them explores the techniques and algorithms used for segmentation, feature extraction, and classification. Most of the previous work does not provide a single place equipped with a comparative analysis of all techniques in blood image analysis. The microscopic examination of blood cells plays a crucial role in diagnosing various haematological disorders such as leukaemia, anaemia, lymphoma, and thalassemia, which can become fatal if not treated timely. Two basic approaches have been adopted for the segmentation and classification of blood cells, *i.e.*, manual analysis or machine-based (automated) analysis. Manual methods are very boring, extensive, exposed to error and prone to haematologist capability. Traditional diagnosis relies heavily on the expertise of haematologists and pathologists, and while automatic analyzers are used, they have limitations, especially in detecting abnormal or neoplastic cells at the pixel level. Manual blood counting requires a tremendous amount of expertise, time, resources, and concentration. Lack of any can cause serious damage in the analysis.

These factors make automated digital image processing essential for the correct and efficient identification of blood-related deformities. Computer-based blood element morphological analysis, like segmentation and classification, are the central elements for machine-based screening of several diseases. Computer-based digital image processing is a very influential tool in the biomedical domain that diminishes human effort, error, and time. Also, it improves judgment precision while saving resources and human resources. These factors highlight the importance of blood cell segmentation and classification as they directly impact the accuracy and efficiency of disease diagnosis. Blood cell segmentation

and classification are important because they enable the detailed analysis of blood smear images, which is essential for identifying and understanding various blood-related disorders. The challenges in this field are diverse, including the following key challenges:

- The segmentation process needs high precision to distinguish different types of blood cells from the background accurately.
- Accurate detection of overlapped structures of blood elements and the classification of these cells into their respective classes.
- The variability of blood elements with respect to shape and size is also another challenge that needs special attention during image processing.
- The presence of multiple artifacts in blood smear images due to any disorder.

The underlying review article explores state-of-the-art works performed previously and a comparative analysis of each method. The key objectives of this study are:

i) Comparative analysis among the techniques is performed for the segmentation, feature extraction, and classification of WBC and RBC.
ii) Identification and categorization of datasets used for blood cell image analysis with respect to usage preference ratio.
iii) We also categorized the features and descriptors that are more important for achieving efficient segmentation and classification based on comparative analysis.
iv) This study also explores the evaluation parameters that are the most commonly used for authentication of the framework's results, along with their accuracies.
v) Provide basic relationship among image processing algorithms, features extraction, and image enhancing techniques to yield better performance, *i.e.*, the combination of these three outperforms the others.
vi) This study will help the researchers find the best algorithms and frameworks for blood cell segmentation, feature extraction, and classification.

## Who is intended for

Diagnosing diseases is a first and critical step in the medical field because such decisions directly impact patient health and quality of life. Enhancing the current diagnostic systems is thus an imperative goal that needs contributions, expert opinion, and decisions from many disciplines. This study is planned for a diverse audience, including experts from the immediate fields like machine learning experts, AI researchers, data scientists, medical image processing domain, and researchers from interdisciplinary backgrounds, *i.e.*, hematologists, pathologists, biomedical engineers, and diagnostic technicians in laboratories. The objective is to provide a comprehensive understanding that can facilitate disease diagnosis advancements in medical image processing, ultimately assisting clinicians and patients. Following is the list of whom the research for:

**Machine learning and AI researcher:** For those working in machine learning, deep learning, and AI-based techniques in healthcare, this survey offers a thorough review of

methodologies, deep learning algorithms, evaluation matrices, and datasets employed for segmentation and classification, thereby guiding future research directions. Applying the surveyed algorithms to the current disease diagnostic system could improve disease detection microscopic imaging performance.

**Medical imaging**: By examining various image-based datasets and methods employed, the survey offers an understanding that can assist in standardizing imaging protocols and techniques for pixel-level analysis. This standardization will help to improve the reliability and reproducibility of medical imaging in hematological diagnostics.

**Hematologist and pathologist:** This survey provides a comprehensive look at the developments in blood cell segmentation and classification through CAD techniques for clinicians specializing in blood disorders. It can guide them to adopt more efficient diagnostic strategies. Identifying the most promising methodologies and technologies can guide healthcare decision-makers on where to allocate resources for maximal impact to perform pixel-level analysis on blood cells.

**Biomedical engineers:** Those involved in creating medical imaging technologies and developing healthcare applications can gain an understanding of the algorithms and machine learning models that have proven effective performance in blood cell analysis.

**Interdisciplinary collaboration:** By identifying gaps in the current state-of-the-art research and raising questions for future research, the survey opens the door for collaboration between medical imaging experts and professionals in machine learning, data science, and hematology.

The rest of the article is organized into the following sections: "Literature Review" describes the literature on the segmentation, feature extraction, and classification of WBCs and RBCs. "Analysis and Discussion" contains an analysis and discussion of all previous work, concluding remarks, and future perspectives discussed in "Ethical Considerations in Automated Diagnosis and Patient Data Privacy".

# LITERATURE REVIEW

In this section, we review the state-of-the-art techniques for WBC and RBC segmentation and classification. The "Survey Methodology" and "Segmentation" sections present the review methodology, which was followed to conduct a systematic review, and detail the WBC/RBC segmentation techniques, respectively. The "Feature Extraction" section describes the features used for segmentation and classification of WBC and RBC. Finally, the "Classification" section describes the tools and techniques used for the classification of WBC and RBC.

## Survey methodology

The following approaches were adopted for searching the most relevant research work for WBC and RBC segmentation and classification.

**Literature search strategy:** A multi-step, systematic approach was employed to search and select articles to ensure comprehensive and unbiased coverage of the literature. The initial search was conducted with the help of Google Scholar to ensure a broad coverage of available literature.

**Keywords:** Popular keywords were chosen from the relevant field to capture the most relevant research work related to blood cell segmentation and classification. The keywords included: "white blood cell," "red blood cell," "disease," "machine learning," "deep learning," "image processing," "blood cell datasets," and "morphological feature of blood cells". These keywords were selected to encapsulate the various dimensions, like the biological and computational context of blood cell segmentation and classification.

**Search refinement:** The keywords were re-arranged and combined in various ways to make the search more precise to the research domain. Only articles written in English were considered to ensure consistency in understanding and explanation.

**Selection criteria:** After obtaining a large list of articles, their abstracts were read carefully to determine their relevancy and contribution to the WBC and RBC analysis. Criteria for selection included innovation, methodological consistency, relevance to the WBC and RBC analysis, and significant contribution to the existing body of knowledge.

**Bias mitigation:** To mitigate any biases, the search and selection process aimed to be as inclusive as possible within the boundaries of relevance and quality. The methodology of each shortlisted article was critically examined to ascertain its scientific validity. This was done to ensure the literature review reflects a balanced view of the existing research, avoiding overrepresenting any particular methodology or viewpoint.

**Finalization:** After this systematic process, a final list of articles was curated for in-depth review and analysis, providing a comprehensive and unbiased survey of blood cell segmentation and classification techniques.

## Segmentation

In the image processing domain, segmentation is a special practice used for partitioning an image into a set of objects (*Shahin et al., 2018*). The basic aim of this practice in the medical image domain is the division of blood cells based on the region of interest (ROI) (*Puigví & Alférez, 2018*). The core step in medical image processing is segmentation, while feature extraction and classification are cited within the machine learning domain (*Rodellar et al., 2018*). The segmentation process primarily aims to divide image elements without overlapping based on ROI. The segmentation process passed through several stages, *i.e.*, 1) microscopic image acquisition, 2) pre-processing for noise removal, 3) image enhancement to get the quality image, and 4) image segmentation (*Shahin et al., 2018*), shown in Fig. 1.

The medical image processing domain's key elements are white blood cells (WBCs) and red blood cells (RBCs). Accurate separation of these cell elements from the background is essential for the rest of the analysis. The following section describes the literature relevant to the WBC and RBC segmentation techniques.

### WBC segmentation

In medical image analysis, segmenting the region of interest (ROI) is the initial rudimentary step for image processing (*Caicedo et al., 2019*). Based on the efficient segmentation of blood cell elements like WBCs, RBCs, and platelets, several analyses can be carried out after the segmentation, *i.e.*, feature extraction, classification, *etc*. These analyses provide valuable

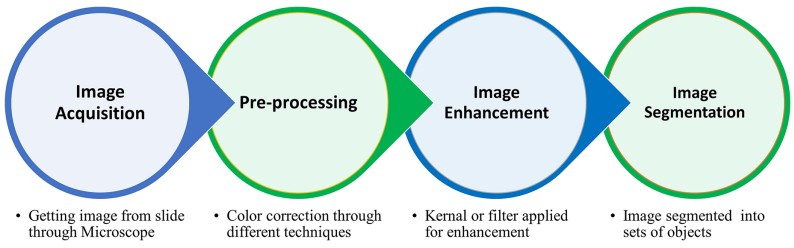

**Figure 1** The complete procedure of the blood cell segmentation process from blood collection to segmentation of ROI.               

information during morphological analysis and disease diagnostic procedures (*Liu & Long, 2019*). In *Nikitaev et al. (2018)* introduced a technique for segmenting leukocytes from blood and bone marrow cells. A total of 1,018 cell images were formed out of 50 samples. Background and erythrocytes (RBCs) are separated using luminance histogram analysis of the color component. Object selection (filling), screening artifact, and identification of "sticky" WBCs were achieved by Watersheds and distance transformation techniques. A total of 82% of the nuclei were allocated correctly. They selected three types of artifacts; the first type of artifact was allocated in 8% of the cells, the second type of artifact was 4%, and the third type of artifact was 12% of the cells. In *Alomari et al. (2014)*, *Monteiro et al. (2020)*, *Miao & Xiao (2018)*, proposed algorithms based on marker-controlled watershed and thresholding, morphological-based techniques for systematically segmenting leukocytes and erythrocytes. The algorithms work in two steps: cell nucleus segmentation and blood cell segmentation. The time complexity of the proposed method is 0.32 s, which is less than Wei and Cao's (0.46 s) method (*Wei & Cao, 2016*). While it is greater than (*Cuevas et al., 2013*), whose method complexity was 0.3 s. The proposed method is consistent and coherent in datasets, with the accuracy rates of the WBC segmentation at 97.2% and RBCs at 94.8%. The proposed method may perform negatively on the poor image quality. *Shirazi et al. (2016a)* proposed a novel technique to enhance and segment leukocytes from microscopic images. Curvelet, transform, and wiener filters were used for image enhancement and noise removal. The snake algorithm and Gram-Schmidt orthogonalization were used for boundary detection and segmentation. *Shirazi et al. (2016c)*, the authors try to overcome the overlapping problems of the blood cells. They got a segmentation accuracy rate of 100%, 96.15%, 92.30%, 92.30%, and 96.15% for basophil, eosinophil, monocyte, lymphocyte, and neutrophil, respectively. They achieved the segmentation by using a Wiener filter and Curvelet transform for image enhancement and noise elimination. They used the ALL-IDB blood cell dataset (*Labati, Piuri & Scotti, 2011*). *Cao, Liu & Song (2018)* performed a deep study on the segmentation of leukocytes in peripheral blood by introducing an algorithm SWAM&IVFS (stepwise averaging method) with consideration of RGB color space, HIS color space, and linear combination of G, H, and S component. Results show that the accuracy rate for segmentation was 93.75% with a standard deviation of 0.5984%. A total of 203 reference images were considered, which were sketched by the medical expert. Of these, 10.34% were monocytes, 24.14% lymphocytes, 60.59% neutrophils, 3.94% eosinophils, and 0.99% basophils. In *Liu et al. (2015)*, the

authors used the nucleus mark watershed operation and mean shift clustering technique to efficiently segment WBCs in peripheral blood and bone marrow images. The whole method is divided into four steps. In the first step, they obtain the nucleus as the inside seed by using RGB and HIS color space. In the subsequent stage, they generated white blood cells (WBCs) as an external seed by employing mean shift clustering operations, extracting the C channel component in the CMYK model, implementing illumination intensity adjustment, and utilizing image enhancement techniques. The third step relates to cell adhesion by using NMWO, followed by post-processing techniques to obtain the nucleus and WBCs. ALL-IDB1 datasets were used for testing the results. The proposed algorithm outperforms in the case of P, R, and F1 measures with an accuracy rate of 99%, 95.5%, and 97%, respectively. *Quiñones et al. (2018)* introduce a novel technique for the WBCs segmentation and counting using the HSV saturation component with blob analysis. A total of 12 images were used for the experiments. This model's accuracy for counting leukocytes was 90.91%, the lowest, and 100%, the highest. While the execution time was 0.032 s slowest and 0.112 s as the highest execution time. *Tosta et al. (2015)* proposed an unsupervised segmentation method for the nuclear structure in WBCs. They used the threshold Neighborhood Valley-emphasis algorithm to select the region of interest in the image. A total of 367 images were used to test the proposed methodology. The accuracy with Jaccard was 89.89%, while with accuracy matrices, 99.57%. *Shahin et al. (2018)* used an adoptive neutrosophic similarity score to segment white blood cells. They proposed an algorithm based on the multi-scale similarity measure in the neutrosophic domain. They used a color segmentation framework for nucleus and cytoplasm segmentation. The quantitative results show high accuracy rates of the segmentation performance measurement 96.5% and 97.2% of the proposed method. In *Abdulhay et al. (2018)*, *Saleem et al. (2022)*, *Sharma & Buksh (2019)* authors proposed leukocyte identification based on the ROI segmentation approach with 98.5% and 95.3% accuracy for detecting leukemia cells using Hybridge CNN and SVM, respectively. Several survey works have been carried out, *Aldrin Karunaharan & Prakash (2018)*, *Bagasjvara et al. (2017)*, *Reyes, Rozo & Morales (2015)*, *Rodellar et al. (2018)*, *Shafique & Tehsin (2018b)*, but most of them target few areas, *i.e.*, comparison of just two techniques for segmentation (Threshold Segmentation Based on Mathematical Morphology (TSMM) and Segmentation Based on Fuzzy Cellular Neural Networks (FCNN)), explore the process of segmentation with different techniques, review on the leukemia detection techniques and detection of acute lymphoblastic leukemia (ALL) cells respectively, but all of them mostly covered traditional approaches not deep learning.

Table 1 is the literature summary regarding developing the WBCs segmentation techniques dataset used for WBC segmentation. It also shows the relevant features and image enhancement techniques commonly used before the WBCs segmentation process.

Table 2 summarizes the literature regarding WBC segmentation performance accuracy and relevant evaluation parameters that are most commonly used for the WBCs segmentation process.

**Table 1 Techniques and datasets used for WBC segmentation.** The WBC segmentation techniques dataset used for WBC segmentation, along with the relevant features and image enhancement techniques commonly applied before the WBCs segmentation process.

| Work | Database | Segmentation | Features | Enhancement | Classifier |
|---|---|---|---|---|---|
| Sharma & Buksh (2019) | ALL-IDB1 108 images | Hybrid segmentation using ALL-DC model | Morphological feature | Firefly optimization algorithm (FOA) based lighting technique | Fuzzy based CNN |
| Al-jaboriy et al. (2019) | ALL-IDB1 108 images | ROI-based segmentation using local pixel information | 4-moment statistical features | Nil | ANN |
| Abdulhay et al. (2018) | 100 microscope images | ROI and edge detection | Local binary patterns and texture features | Median filter and adaptive weighted median filter, | Support vector machine |
| Shahin et al. (2018) | BS_DB3, ALL_DB1, 2 108 and 260 images | Otsu's thresholding | Nil | Smoothing procedure with average filter | Modified watershed transform |
| Quiñones et al. (2018) | 12 blood smear images | Zak algorithm | Eccentricity and area | HSV color space | Nil |
| Cao, Liu & Song (2018) | 203 and 70853 drown themselves | SWAM&IVFS, fuzzy divergence based | Texture | RGB and HIS color space | Nil |
| Miao & Xiao (2018) | 100 manually collected images | Marker-controlled watershed | Nil | Histogram base technique | Marker-controlled watershed |
| Nikitaev et al. (2018) | 1,018 cells element | Modified watersheds/ distance transformation | Nil | luminance histogram analysis | Supervised learning method |
| Shirazi et al. (2016c) | Nil | Snake algorithm, Gram-Schmidt orthogonalization | Nil | curvelet transform, wiener filter | Nil |
| Di Ruberto, Loddo & Putzu (2016) | ALL-IDB, 108 images SMC-IDB 367 images IUMS-IDB 195 images | Segmentation Via SVM | Texture feature | NNS (nearest neighbor search) with Euclidean distance | Nil |
| Liu et al. (2015) | 306 images from hospital | Nucleus mark watershed, Mean shift clustering | Color feature | RGB and HIS color space | Nil |
| Tosta et al. (2015) | 367 images | Neighbourhood valley-emphasis | Nil | A median filter and linear contrast stretching | Nil |
| Shirazi et al. (2016b) | ALL-IDB1 108 images | Thresholding and mathematical morphology | Geometrical and texture | Wiener filter and curvelet transform | Back-propagation neural network |
| Alomari et al. (2014) | 100 microscopy images | Thresholding | Color feature | Nil | Nil |

Figure 2, Part B, shows the literature regarding WBC segmentation. It shows the research work, techniques used, and common limitations in the domain of WBC segmentation.

### RBC segmentation

In Das, Maiti & Chakraborty (2018), authors put forward a method for the automated detection of nucleated red blood cells (RBCs) in peripheral blood smears. A Special Fuzzy C-mean algorithm was used for image enhancement. They got an accuracy of 99.42% for

**Table 2 Comparative analysis of segmentation techniques and their performance used for WBC image analysis.** A summary of various segmentation methods used by different authors, along with the corresponding performance accuracy achieved and the evaluation parameters used to measure the effectiveness of each technique.

| Author | Segmentation | Performance accuracy | Evaluation parameter |
|---|---|---|---|
| *Sharma & Buksh (2019)* | Hybrid segmentation using ALL-DC model | 98.02%, 70.07%, and 86.2% | Precision, recall, and F1-measure |
| *Al-jaboriy et al. (2019)* | ROI-based segmentation using local pixel information | 97% | Accuracy |
| *Abdulhay et al. (2018)* | ROI and edge detection | 95.3% and 91.66% | Accuracy, specificity, and sensitivity |
| *Shahin et al. (2018)* | Otsu's thresholding | 97.6% | Performance measurement |
| *Quiñones et al. (2018)* | Zak algorithm | 98.88% | Counting accuracy |
| *Cao, Liu & Song (2018)* | SWAM&IVFS, fuzzy divergence based | 93.75% | Accuracy |
| *Miao & Xiao (2018)* | Marker-controlled watershed | 97.2% and 94.8% | Over/under-segmentation and fault rate |
| *Nikitaev et al. (2018)* | Modified watersheds/distance transformation | 82% | Accuracy |
| *Di Ruberto, Loddo & Putzu (2016)* | Segmentation *Via* SVM | 99.73% | Counting accuracy |
| *Liu et al. (2015)* | Nucleus mark watershed, mean shift clustering | 99%, 95.5%, and 97% | Precision, recall, and F1 score |
| *Tosta et al. (2015)* | Neighborhood valley-emphasis | 89.89% and 99.75% | Jaccard's similarity coefficient and accuracy |
| *Shirazi et al. (2016b)* | Thresholding and mathematical morphology | 96.15% | Accuracy |
| *Alomari et al. (2014)* | Thresholding | 98.4% | Precision, recall, and *F*-measurements |

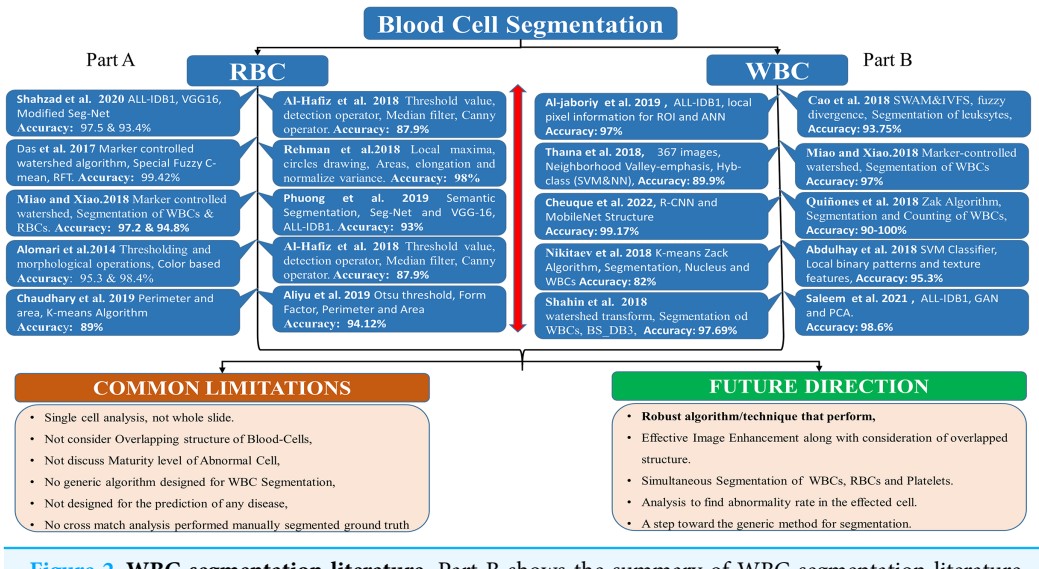

**Figure 2 WBC segmentation literature.** Part B shows the summary of WBC segmentation literature work, techniques, accuracy, and common limitations.

detecting nucleated RBCs from the blood smear images. A multilevel threshold approach made the selection of nucleated blood cells. Segmentation was achieved by using a marker-controlled watershed algorithm. In *Miao & Xiao (2018)*, *Shahzad et al. (2020)*, the authors used a marker-controlled watershed and a modified SegNet model. They accomplished

simultaneous segmentation of red blood cells (RBCs) and white blood cells (WBCs) with accuracies of 94.8% 97.2%, and 97.45% and 93.43%, respectively. In *Shahzad et al. (2021)*, *Shirazi et al. (2016b)*, the author proposed a novel technique for automated diagnosis of several blood diseases like AIDS, iron deficiency, blood disorders, platelets, malaria, leukemia and anemia. The author proposed machine-based blood morphological analysis. The author performed the segmentation of RBCs and WBCs along with their sub-types. They merged different methodologies like feature, color domain, and classifier for cell segmentation. In *Imran & Ahmad (2017)*, *Shirazi et al. (2016a)* devised a statistical thresholding approach for segmenting red blood cells (RBCs), which was then succeeded by using Fuzzy C-means for accurate segmentation and boundary detection. Texture and geometrical features were used for classification. They tested their results on the ALL-IDB blood cell dataset. They performed single-cell-based RBC classification. A total of 10,000 RBCs were used, of which 8,000 were normal and 2,000 were abnormal. The overall classification accuracy of this technique was 96%. Another work (*Al-Hafiz, Al-Megren & Kurdi, 2018*) has been performed on RBC segmentation using thresholding value and corner detection with 87.9% accuracy. This method fails to accurately detect edges of images that show overlapped structures of cells. In *Rehman et al. (2018)*, the author introduced an innovative technique for separating Rouleaux red blood cells from thin blood smears by implementing distance transform and local maxima. The proposed method was assessed using three evaluation parameters, namely TPR (96%), ER (4%), and AC (98%).

Furthermore, semantic segmentation was conducted by *Chaudhary et al. (2019)*, *Tran et al. (2019)* using SegNet architecture and VGG, respectively, for the segmentation and quantification of white blood cells (WBCs) and red blood cells (RBCs) in the ALL-IDB-I dataset. This architecture achieved an average accuracy of 93%. In *Aliyu et al. (2019)*, normal and abnormal RBCs were identified by leveraging Form Factor, Perimeter, and area features, achieving an accuracy of 94%. However, the method was ineffective when dealing with noisy images.

Table 3 shows the literature summary regarding developing segmentation techniques and the dataset used for RBC segmentation. It also shows the relevant features and image enhancement techniques commonly used for RBC segmentation.

Table 4 summarizes the literature regarding RBC segmentation performance accuracy and relevant evaluation parameters that are most commonly used in the literature.

Figure 2, part A summarizes the literature regarding RBC segmentation. It shows the research work, techniques used, and common limitations in the domain of RBC segmentation.

## Feature extraction

An identification process in which a set of quantitative descriptors are identified for each ROI division is called feature extraction (*Rodellar et al., 2018*). The core steps (*Puigví & Alférez, 2018*) involved in image analysis are shown in Fig. 3.

After segmentation, the next step is feature extraction (*Mohammed et al., 2013*; *Alférez et al., 2015*), which is critical for efficient classification. Quantitative descriptors are usually

**Table 3  Techniques and datasets used for RBC segmentation.** The WBC segmentation techniques dataset used for WBC segmentation, along with the relevant features and image enhancement techniques commonly applied before the WBCs segmentation process.

| Work | Database | Segmentation | Features | Enhancement | Classifier |
|---|---|---|---|---|---|
| Aliyu et al. (2019) | Online from ash bank and shutter stock | Otsu threshold and Binarization | Form factor, perimeter, and area | Nil | SVM |
| Chaudhary et al. (2019) | Nil | K-means algorithm | Perimeter and area | Nil | Nil |
| Tran et al. (2019) | ALL-IDB1 108 images | SegNet and VGG-16 | VGG-16 | Distance transform | Nil |
| Miao & Xiao (2018) | 100 manually collected images | Marker-controlled watershed | Nil | Histogram base technique | Marker-controlled watershed |
| Al-Hafiz, Al-Megren & Kurdi (2018) | BBBC dataset | Threshold value using detection operator | Gradient magnitude and canny operator | Median filter | Nil |
| Rehman et al. (2018) | Thin blood smear images | Local maxima, circles drawing | Areas, elongation and normalize variance | Thresholding for binary conversion | Nil |
| Das, Maiti & Chakraborty (2018) | 950 nucleated blood cells | Marker-controlled watershed algorithm | Color feature | Special Fuzzy C-mean | RFT |
| Imran & Ahmad (2017) | ALL-IDB1 108 images | Fuzzy C-means | Texture and geometrical | Nil | SVM |
| Shirazi et al. (2016c) | Nil | Snake algorithm, Ostu Thresholding | Color feature | HSV color | SVM |
| Alomari et al. (2014) | 100 microscopy images | Thresholding | Color feature | Nil | Nil |

**Table 4  Evaluation metrics for RBC segmentation techniques.** A detailed overview of the segmentation methods used for the segmentation of red blood cells (RBCs). It also illustrates the performance accuracy obtained from each method.

| Work | Segmentation | Performance accuracy | Evaluation parameter |
|---|---|---|---|
| Aliyu et al. (2019) | Otsu threshold and binarization | 94.12% | Accuracy, specificity, and sensitivity |
| Chaudhary et al. (2019) | K-means algorithm | 89% | Accuracy |
| Tran et al. (2019) | SegNet and VGG-16 | 93% | Accuracy, boundary F1 |
| Miao & Xiao (2018) | Marker-controlled watershed | 97.2% and 94.8% | Over/under-segmentation and fault rate |
| Al-Hafiz, Al-Megren & Kurdi (2018) | Threshold value using detection operator | 87.9% | Sensitivity, precision, and F1-Score |
| Rehman et al. (2018) | Local maxima, circles drawing | 96%, 98%, and 4% | TPR, accuracy, ER, and TNR |
| Das, Maiti & Chakraborty (2018) | Marker-controlled watershed algorithm | 99.42% | Accuracy |
| Shirazi et al. (2016c) | Snake algorithm, ostu thresholding | 96% | Accuracy |
| Alomari et al. (2014) | Thresholding | 98.4% | Precision, Recall, and F-measurements |

divided into geometric, morphological, and color, and texture features. In the medical image processing domain, efficient detection of WBCs and RBCs is correlated with these features. The categories of feature extraction are shown in Fig. 4. The description of the quantitative descriptor is given below.

### Geometric or morphological features

Morphological features directly interact with visual reflection and narrative made by the pathologist. Usually, these features are easy to interpret, observe, and segment for different diagnostic analyses. The most commonly used descriptor parameters are

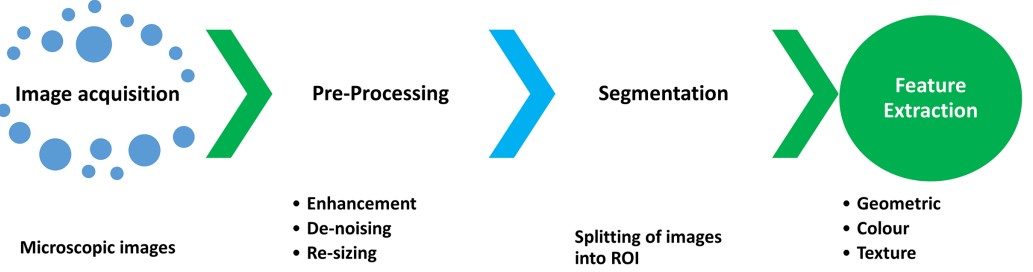

**Figure 3 The steps involved in the feature extraction process during the segmentation and classification of WBCs and RBCs.**

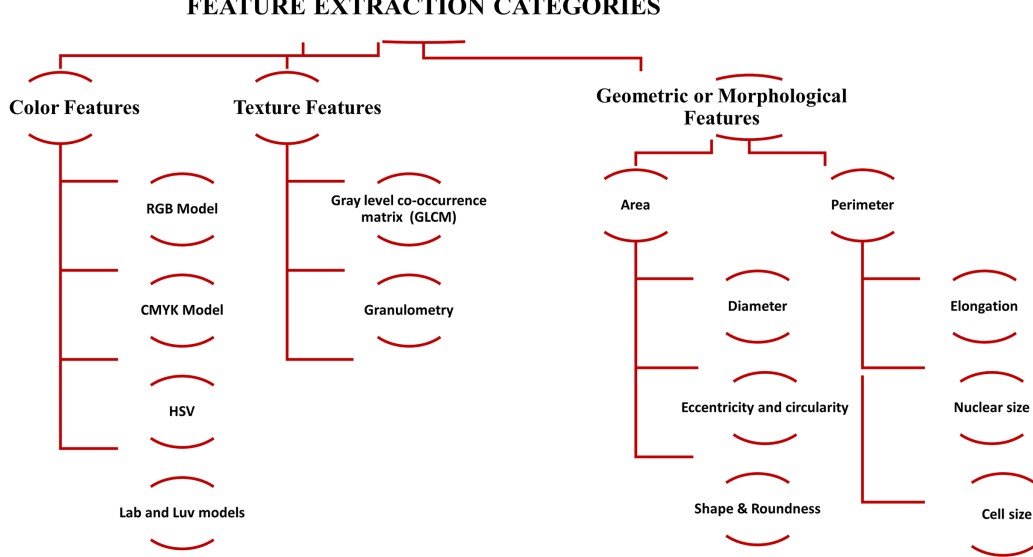

**Figure 4 The summary of feature extraction categories, *i.e.*, color features, texture features, and geometric or morphological features used for blood cell classification.**

(*Puigví & Alférez, 2018*) area, perimeter, diameter, elongation, convexity, nuclear size, cell size, shape, nucleus-cytoplasmic ratio, nuclear eccentricity, roundness, eccentricity, and circularity.

## Color features

Color-based image descriptors also play a crucial role in identifying the ROI and edges of the object. It is a physical property commonly used for visual observation. Various models have been used for color feature extraction, *i.e.*, RGB, CMYK, HSV, and Lab and Luv models (*Elhassan et al., 2022*; *Rodellar et al., 2018*). These models perform work based on a key tool called a histogram.

## Texture features

Despite the color and geometric features, texture descriptors are relatively difficult to observe for visual interpretation. In the digital image processing domain, the texture is

**Table 5 Feature for image enhancement.** The diversity of blood feature selection used in recent studies. It also highlights the high accuracy percentages achieved by various classifiers and the effectiveness of the selected features in blood cell segmentation and classification tasks, which produces outstanding results.

| Work | Features | Classifier | Database | Accuracy % |
|---|---|---|---|---|
| *Du et al. (2019)* | Morphological feature | CNN | 17933 samples | 90.7 |
| *Mishra, Majhi & Sa (2019)* | DOST+PCA+LDA, | ADBRF, RF | ALL-IDB1 | 98.6, 99.6 |
| *Liu & Long (2019)* | Morphological feature | Augmented enhanced bagging classifier | 76 images | 85 |
| *Hegde et al. (2019)* | Morphological feature | TissueQuant algorithm | ALL-IDB1 | 96.5 |
| *Sharma & Buksh (2019)* | Morphological feature | fuzzy based CNN | ALL-IDB1 | 98.02 |
| *Al-jaboriy et al. (2019)* | 4-moment statistical features | ANN | ALL-IDB1 | 97 |
| *Agaian, Madhukar & Chronopoulos (2018)* | Cell energy, color features and morphological | SVM | ALL-IDB1 | 97 |
| *Liang et al. (2018)* | Size and intensity of the nucleus | CNN-RNN | BCCD dataset | 94.3 |
| *Hegde et al. (2018)* | Area, perimeter, circulatory, convexity and solidity | Hybrid-classifier (SVM & NN) | 117 images acquired by themselves. | 93.4 |
| *Sampathila, Shet & Basu (2018)* | GLCM | GUI | Leishman-stained thin blood smear slides | 96.7 |
| *Kihm et al. (2018)* | Shape, size | CNN | 4,000 manually classified images | 91.8 |
| *Das, Maiti & Chakraborty (2018)* | Mean intensity, standard deviation, skewness, kurtosis, and entropy | Random Forest (RF) | 950 nucleated blood cells | 99.42 |
| *Mishra, Majhi & Sa (2018)* | GLRLM and texture | SVM | ALL-IDB1 | 96.97 |
| *Xu et al. (2017)* | Shape | Deep CNN | 7,000 single RBCs images | 91.01 |
| *Acharya & Kumar (2017)* | Area, perimeter, diameter, shape and geometric | Modified watershed transform | Nil | 98 |
| *Mishra et al. (2017a)* | GLCM and PCA (nucleus) | RF | ALL-IDB1 | 99.04 |
| *Mishra et al. (2017b)* | Discrete cosine transform (DCT) | SVM–L | ALL-IDB | 89.76 |
| *Singhal & Singh (2016)* | LBP and GLCM | SVM | ALL-IDB1 | 93.8 |
| *Di Ruberto, Loddo & Putzu (2016)* | Morphological, color, and textural | SVM–P | ALL-IDB1, ALL-IDB2 | 93.2 |
| *Rawat et al. (2015)* | Shape and texture features | ANN | ALL-IDB1 | 97.2 |
| *Neoh et al. (2015)* | Color, shape, and texture | Dempster-shafer | ALL-IDB1 | 96.7 |
| *Mohapatra, Patra & Satpathy (2014)* | Morphological, color, and texture | $EOC_5$ | Leishman-stained peripheral blood sample | 99 |
| *Nasir, Mashor & Hassan (2013)* | Size, color, and shape | MLP | 500 manually collected images | 95.70 |

defined with the help of pixel tone, density of pixel, uniformity, and their spatial relation with each other (*Puigví & Alférez, 2018*). There are two main texture descriptor classes: Gray-level co-occurrence matrix and granulometry.

Table 5 shows the quantitative features, relevant accuracy, and the blood database most commonly used for blood cell segmentation and classification.

**Table 6 Overview of datasets and algorithms for WBC classification.** The dataset utilization, segmentation techniques, feature types, enhancements applied, and classification algorithms used in various research works on WBC classification.

| Author | Database | Segmentation | Features | Enhancement | Classification |
|---|---|---|---|---|---|
| *Hegde et al. (2019)* | ALL-IDB1 108 images | TissueQuant algorithm | Morphological feature | Color component of RGB | TissueQuant algorithm |
| *Liang et al. (2018)* | BCCD 1238 images | Nil | Size and intensity of the nucleus | Matrix Transformation | CNN-RNN |
| *Hegde et al. (2018)* | 117 images | TissueQuant | Area, perimeter, circulatory, convexity and solidity | Color contrast technique | Hybrid-classifier (SVM & NN) |
| *Yadav, Zele & Patil (2018)* | Nil | K-means Zack Algorithm | Color feature, geometric feature | Prewitt and Sobel | SVM and ANN |
| *Di Ruberto, Loddo & Putzu (2016)* | ALL-IDBII 260 images, IUMS-IDB 195 images | Pixel-based | Pixel-wise features | RGB channel | SVM |
| *Liu & Long (2019)* | 76 images | Inception ResNets, ImageNet | Nil | Otsu's method and erosion operation | Augmented enhanced bagging ensemble |
| *Vogado et al. (2018)* | ALL-IDB1 108 images | AlexNet + Vgg-f | Transfer learning | Nil | SVM |
| *Othman, Mohammed & Ali (2017)* | Nil | Threshold-based | Shape, intensity and texture | GLCM | MLP-BP neural network |
| *Zhao et al. (2017)* | ALL-IDB1 108 images | Nil | PRICoLBP and PRICoLBP | Nil | Granularity feature and SVM |
| *Agaian, Madhukar & Chronopoulos (2018)* | ALL-IDB1 108 images | K-means clustering algorithm | Morphological features | L * a * b * color space | SVM |

## Classification

### WBCs classification

After completing ROI segmentation and succeeding feature extraction process, the arithmetical descriptors distinctively identify each blood cell element. Classification is a procedure that labels the ROI with these numerical descriptors among predefined classes. This section will explore several research works on classifying blood cell elements using traditional and deep learning approaches. Table 6 shows the literature summary regarding developing classification techniques and datasets used for WBC classification. It also shows the relevant features and image enhancement techniques commonly used before the WBC classification. In *Liang et al. (2018)*, authors proposed a CNN-RNN framework to classify blood elements. They initially introduced RNN and then combined it with CNN for better results. For classification, the proposed methodology integrates the features extracted by the RNN and local features obtained from the CNN. A transfer learning technique transferred pre-trained weight parameters to the CNN section. They used BCCD (*Shenggan, 2018*) (small-scale dataset for blood cell detection) and got 12,444 enhanced blood cells by pre-processing the BCCD dataset, out of which 2,487 were test data and 9,957 were training data with 20% and 80% ratio, respectively. This dataset is divided into four different categories, namely, monocyte (2,478), lymphocyte (2,483), neutrophil (2,499), and eosinophil (2,497) for the training purposes, while in the test dataset, the distribution of cells was 620, 623, 624, and 623 as monocyte, lymphocyte, neutrophil and

eosinophil respectively. The entire network used RGB images of size $320 \times 340 \times 3$ pixels. Results show that CNN-RNN models (Xception-LSTM, Inception V3-LSTM, RestNet50-LSTM, and Xception- RestNet50-LSTM) outperform with a classification accuracy of 90.79%, 87.45%, 89.38%, and 88.58% as compared to the other CNN models (Inception V3, RestNet50, Xception) 84.08%, 87.62%, and 88.70% respectively. But, regardless of high classification accuracy based on the dataset, these experiments were performed on most single-cell images. The single-cell classification process may take 3.8 s, which is clinically too slow. This method also faces problems where multiple cells overlap and needs to design a task-specific classifier to consider such issues. In *Hegde et al. (2018)*, proposed an image-processing algorithm for classifying WBCs and nuclei based on the nuclei features. TissueQuant and image enhancement methods were used to manage color and illumination variation for nuclei detection. They used 117 images with a magnification rate of 100X acquired with an OLYMPUS CX31 microscope with $1,600 \times 1,200$ resolution from Leishman-stained peripheral blood smear. Out of 117 images, 14 were basophils, 30 neutrophils, 33 lymphocytes, 23 monocytes, and 22 eosinophils. Shape and texture features of the detected nuclei were used to classify white blood cells. A five-class approach with a neural network and a cell-by-cell approach with a hybrid classifier (SVM & NN) was used to classify WBCs. By comparing both approaches, cell-by-cell approach outperformed with the rate of 1.4%. The accuracy of other WBCs required more analysis of cytoplasm. This method outperformed the detection of lymphocytes and basophils with 100% accuracy by considering two features, *i.e.*, shape and texture. But the accuracy of neutrophils and eosinophils was 93.4% and 94.3%, and the sensitivity rate was 81.8% and 86%, respectively. More study of cytoplasm is required for the accurate detection of other WBCs. In *Yadav, Zele & Patil (2018)*, introduced an automatic image-handling framework for the early detection of blood cancer. From the microscopic image, they examine shape, size, cell nuclei, and blood cell distribution for the determination of cancer. Prewitt and Sobel or Canny were used to remove artifacts and noise. Segmentation was achieved using region-based segmentation, K-means Zack Algorithm, Morphological operation, gradient magnitude, and watershed transform with a supervised machine learning approach for classification. In *Jung et al. (2022)*, *Othman, Mohammed & Ali (2017)*, authors introduced feed-forward back-propagation neural network-based and generative adversarial networks-based techniques for WBCs classification. They got a 96% classification accuracy rate of white blood cells with 16 selected features. However as they increased the number of features up to 69, the accuracy rate decreased by 89.74%. *Di Ruberto, Loddo & Putzu (2016)* proposed a white blood cell count (WBCC) system to speed up and accurately count cells. WBCs are counted by applying the circular Hough transform, exploiting the grey-level information. This method was tested on ALL-IDB1, ALL-IDB2, Raabin-WBC (*Kouzehkanan et al., 2022*), and IUMS-IDB datasets (*Alomari et al., 2014*). The accuracy rate of counting WBCs was 99.2%. For the segmentation of blood cells, they used the SVM technique. Several deep learning-based techniques (*Agaian, Madhukar & Chronopoulos, 2018*; *Cheuque et al., 2022*; *Liu & Long, 2019*; *Mishra, Majhi & Sa, 2019*; *Rawat et al., 2017*; *Shafique & Tehsin, 2018a*; *Sharma et al., 2022*; *Ullah et al., 2023*; *Vogado et al., 2018*; *Zhao et al., 2017*) have been developed for the detection of leukemic cells in microscopic images.

**Table 7 Performance evaluation metrics for WBC classification.** The performance accuracy percentages and the evaluation parameters such as accuracy, precision, recall, sensitivity, specificity, and F-measures used to assess the effectiveness of various classification algorithms for WBCs.

| Author | Classification | Performance accuracy % | Evaluation parameter |
|---|---|---|---|
| *Hegde et al. (2019)* | TissueQuant algorithm | 96.5 | Accuracy, precision and recall |
| *Liang et al. (2018)* | CNN-RNN | 90.79 | Accuracy |
| *Hegde et al. (2018)* | Hybrid-classifier (SVM & NN) | 86 and 95 | Sensitivity and accuracy |
| *Di Ruberto, Loddo & Putzu (2016)* | SVM | 99.73 | Accuracy |
| *Liu & Long (2019)* | Augmented enhanced bagging ensemble | 84, 85 and 84 | Precision, recall and F1 score |
| *Vogado et al. (2018)* | SVM | 99 | Precision, recall, accuracy and kappa index |
| *Othman, Mohammed & Ali (2017)* | MLP-BP neural network | 96 | Accuracy |
| *Zhao et al. (2017)* | Granularity feature and SVM | 85.3 and 97.1 | Sensitivity and precision |
| *Agaian, Madhukar & Chronopoulos (2018)* | SVM | 98.5, 97.8, 95.7 and 97.1 | Precision, specificity, sensitivity and F-measures |

Most of them classified the WBCs into their subtypes with heterogeneous accuracy rates shown in Table 7. Nuclei detection of white blood cells was demonstrated in *Hegde et al. (2019)* using a novel deep learning framework based on the TissueQuant algorithm with 99% and 96.5% average accuracy and dice coefficient, respectively.

Table 7 shows the summary of previous work regarding WBC classification performance accuracy. 3rd column shows the evaluation parameters used for the evaluation of WBC classification.

Figure 5, Part A summarizes the whole literature regarding RBC classification, while Part B, summarizes the literature regarding WBC classification. It shows the research work, techniques, and common limitations in the WBC and RBC classification domains.

### RBCs classification

In *Kihm et al. (2018)*, authors classify RBCs on the basis of their shape. They introduced an outlier tolerant machine learning technique to find out the slipper-shaped and croissant-shaped cells in the blood. They also find the transition point between both stable and unstable phases. Results show that this technique classifies the croissant-shaped cell with an accuracy rate of 85.6% and the slipper-shaped cell with a 91.8% accuracy rate. *Shirazi et al. (2017)*, *Imran & Ahmad (2017)* introduced a novel technique based on the extreme learning machine (ELM) approach for classifying RBC images. Statistical-based thresholding methods were used for initiating RBCs segmentation, followed by the fuzzy C-means for efficient segmentation and boundary detection. Texture and geometrical features were used for classification. They tested their results on the ALL-IDB blood cell dataset. They performed single-cell-based RBC classification. A total of 10,000 RBCs were used, of which 8,000 were normal and 2,000 were abnormal. The overall classification accuracy of this technique was 96%. Another work (*Das, Maiti & Chakraborty, 2018*) has been carried out for the classification of nucleated RBCs by using 250 blood smear images. This technique is based on

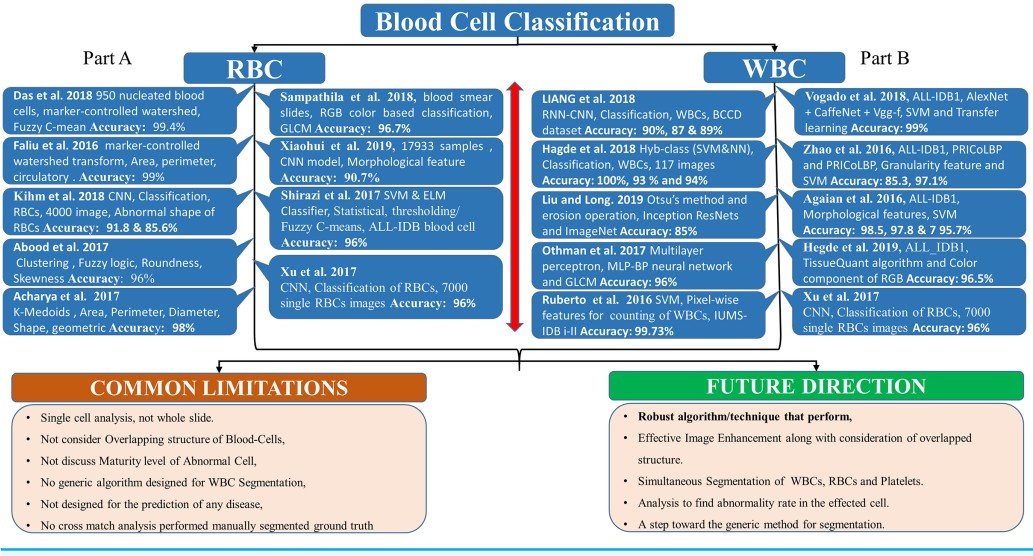

**Figure 5 Blood cell classification literature.** (A) The summary of RBC classification literature work, techniques, accuracy, and common limitations. (B) The summary of WBC classification.

the fuzzy C-mean algorithm with an accuracy rate of 99.42%. *Yi, Moon & Javidi (2016)* Proposed a technique for automatically classifying RBCs by selecting a linear or non-linear classifier. This method used Gabor-filtered holographic images to measure covariance matrices to classify RBCs. The methodology consists of three basic steps. 1) Numeric reconstruction of RBCs from the hologram, 2) marker-controlled watershed transform algorithm was used to segment RBCs, and 3) Gabor wavelet transform was applied for classification along with multivariate statistical test to evaluate the equality of the covariance of matrices. *Abood, Karam & Hluot (2017)* proposed a technique for classifying RBCs disease using the Fuzzy logic theory. They performed statistical-based analyses like mean, standard deviation, variance, roundness, skewness, and kurtosis, considering size, shape, and color features. The proposed method got an accuracy of 98% for red blood cells. In *Acharya & Kumar (2017)*, the authors presented a computer-aided system designed to diagnose blood disorders, such as Anemia, by classifying red blood cells (RBCs). The K-medoids algorithm was employed to extract white blood cells (WBCs) from the image. To differentiate RBCs from WBCs, granulometric analysis was performed on 1,000 blood smear images obtained from Kasturba Medical College, Manipal, Karnataka, and *Atlas of Hematology (2023)*, Hemato-pathology laboratory (*Mathew et al., 2015*), and the Internet. The system yielded a 98% accuracy rate for correctly identifying Anemia; however, the execution time increased with the number of RBCs present. *Xu et al. (2017)* proposed a deep convolutional neural network (CNN) for classifying RBCs in sickle cell anemia. They introduced a high-throughput framework consisting of three stages. First, they developed an automatic hierarchical RBC extraction method for detecting RBCs. Second, they applied a mask-based RBC patch-size normalization method to normalize the segmented single RBCs areas into uniform sizes. Third, they employed deep CNNs to classify the RBCs. A total of eight sickle

**Table 8 The techniques and datasets used for RBC classification.**

| Author | Database | Segmentation | Features | Enhancement | Classification |
|---|---|---|---|---|---|
| *Du et al. (2019)* | 17,933 samples from the hospital | ImageNet model | Morphological feature | Nil | CNN model |
| *Sampathila, Shet & Basu (2018)* | Leishman-stained thin blood smear slides | RGB color based | GLCM | Color space | GUI |
| *Kihm et al. (2018)* | 4,000 Manually classified images | Nil | Shape, size | By convolution of NN | CNN |
| *Imran & Ahmad (2017)* | ALL-IDB1 108 images | Statistical based thresholding | Morphological | Rayleigh distribution | SVM and ELM |
| *Das, Maiti & Chakraborty (2018)* | 950 blood cells | Marker-controlled watershed algorithm | Mean intensity, standard deviation, skewness, kurtosis and entropy | Special Fuzzy C-mean | Random forest |
| *Yi, Moon & Javidi (2016)* | 117 images Manual collection | Marker-controlled watershed transform | Area, perimeter, circulatory *etc.* | Watershed transform algorithm | Gabor-filtered holographic |
| *Abood, Karam & Hluot (2017)* | Nil | Clustering | Shape and color | | Fuzzy logic |
| *Xu et al. (2017)* | 7,000 single RBCs | Nil | Shape | Geometric transformations | Deep CNN |
| *Acharya & Kumar (2017)* | 1,000 images manually collected | K-medoids algorithm | Area, perimeter, diameter, shape, geometric | Nil | Modified watershed transform |

cell disease patients were selected for collecting 7,000 single RBC images. The proposed method got a 91.01% mean accuracy for RBCs classification under different folds with a mean evolution accuracy of 89.28%. *Du et al. (2019)*, *Sampathila, Shet & Basu (2018)* proposed a framework for the classification of malaria parasites using RBCs morphological features. The later referred technique uses convolutional neural network for automatic classification blood cell that have ability to integrate with facial detection system.

Table 8 illustrate the literature concerning the development of classification techniques and dataset used for RBC classification. It also shows the features and image enhancement techniques commonly used for RBC classification.

Table 9 shows the summary of previous work regarding RBC classification performance accuracy. A total of 3rd column shows the evaluation parameters used for the evaluation of RBC classification.

## ANALYSIS AND DISCUSSION

This work explains the basic outlines regarding machine learning techniques and datasets associated with a specific domain of medical image processing, *i.e.*, blood cell segmentation and classification using microscopic images of WBCs and RBCs. To the best of our knowledge, this is the first comprehensive review in the blood cell image analysis domain that approximately grabs all the techniques and architecture used for blood cell segmentation and classification. We have systematically discussed all the machine learning techniques that have been used in previous works to find out their implementation logic

**Table 9 Performance evaluation parameters used for RBC classification.**

| Author | Classification | Performance accuracy % | Evaluation parameter |
|---|---|---|---|
| *Du et al. (2019)* | CNN model | 90.7 | Precision, recall and F1 score |
| *Sampathila, Shet & Basu (2018)* | GUI | 96.7 | Accuracy |
| *Kihm et al. (2018)* | CNN | 85.6 & 91.8 | Prediction accuracy |
| *Imran & Ahmad (2017)* | SVM and ELM | 96 | Accuracy |
| *Das, Maiti & Chakraborty (2018)* | Random forest | 99.42 | Accuracy |
| *Yi, Moon & Javidi (2016)* | Gabor-filtered holographic | 99 | Accuracy |
| *Abood, Karam & Hluot (2017)* | Fuzzy logic | 98 | Accuracy |
| *Xu et al. (2017)* | Deep CNN | 91.01, 89.28 | Accuracy, mean evaluation accuracy |
| *Acharya & Kumar (2017)* | Modified watershed transform | 98 | Accuracy |

along with their pros and cons. Literature comprises three main sections: 1) segmentation, 2) feature extraction, and 3) classification. Each section is subdivided into a textual description of relevant techniques and a tabulated and diagrammatic summary. A brief comparative analysis of each section is given below:

## Analysis of WBC and RBC segmentation

The literature reveals that 31% of the research work concerning WBC segmentation used ALL-IDBI-II. Out of them, *Al-jaboriy et al. (2019)*, *Sharma & Buksh (2019)*, *Shirazi et al. (2016b)* extract morphological features for segmentation and attain accuracies of 96.15%, 98.02%, and 97%, along with accuracy, precision, recall, and F1-measure as evaluation parameters, respectively, shown in Tables 1 and 2. Figure 6 shows that 18% of the WBC segmentation work used UMS-IDB, SMC-IDB, and BS_DB3 (6% each) datasets, while 50% of researchers collected images manually only for their work. While in the case of RBC segmentation, 20% of researchers chose the ash bank (*Saki et al., 2024*) and BBBC dataset (*Mooney, 2023*) (10% each). 60% of the researchers refused to use predefined datasets and obtained images from the manual collection. The rest of the 20% selected the ALL-IDB dataset for RBC segmentation. To obtain better WBC and RBC segmentation results, 80% of the research extracted texture and morphological features (40% each), while only 20% extracted color features Fig. 7. While in the case of RBC segmentation, 60% of researchers used texture and color features (30% each), respectively, while the rest of the 40% of the work used morphological features. Table 2 shows the comparative analysis among WBC segmentation techniques alongside their evaluation parameters with performance accuracies (in percent).

## Discussion on feature extraction of blood elements

Proper feature extraction of an image is the key to the segmentation and classification process. There are three main classes of image features: i) texture features, ii) color features, and iii) geometrical or morphological features. Figure 8 shows that 55% of the previous research on classification purposes used morphological features, while 28% and 17% used texture and color features.

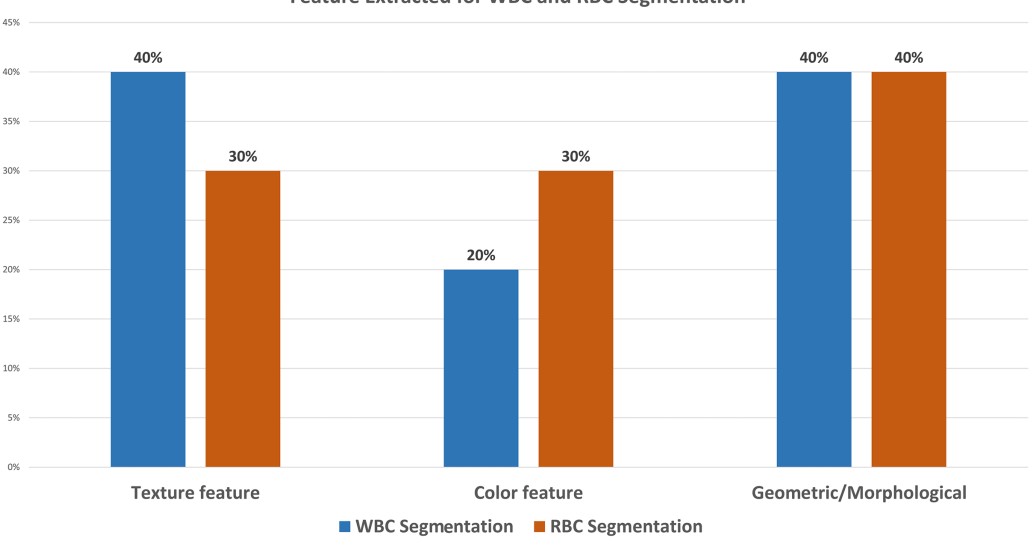

**Figure 6 The comparative analysis of blood cell dataset selection for WBC and RBC segmentation purposes.** These illustrate that most of the researchers choose manual collection rather than using predefined medical image datasets.

**Figure 7 The ratio of blood cell feature selection for analysis for WBC and RBC segmentation.** The results were extracted from the reviewed literature. This shows that texture features and geometric or morphological features got more attention as compared to the color features.

## Analysis of WBC and RBC classification

Table 5 demonstrates that the classification accuracy rate with morphological features alongside ALL-IDB dataset is high compared to the color and texture features. Tables 6 and 8 show the techniques and datasets used for WBC and RBC classification, respectively.

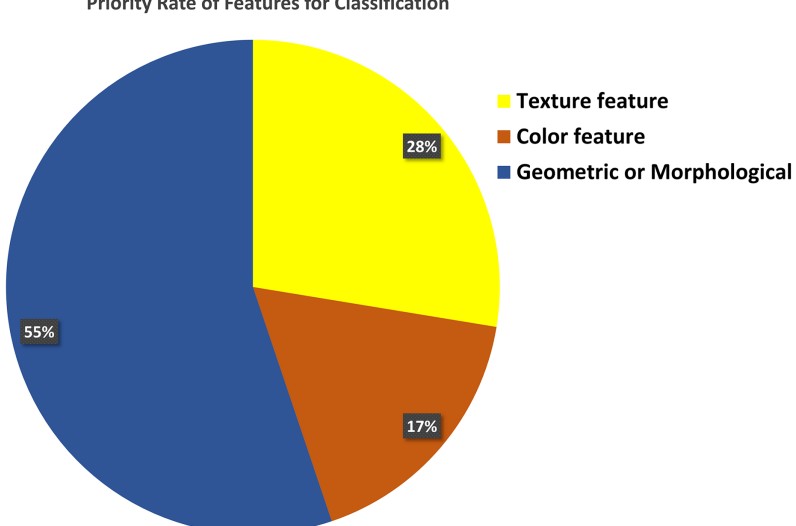

**Figure 8 Feature selection for classification.** Most of the researcher consider geometric or morphological features for the classification of blood cell elements.

Literature reveals that ALL-IDB is the most favorable dataset, *i.e.*, 45% of researchers used this for WBC classification purposes. In contrast, researchers prioritize manually collecting (73%) blood images for RBC classification, as shown in Fig. 9. The rest of the 27% literature work used the ALL-IDB dataset for RBC classification. Figure 10 illustrates that for WBC and RBC classification, researchers prioritize morphological features before the classification, *i.e.*, 55% and 80%, respectively. The color feature is the 2[nd] choice for WBC classification, while texture features are the 2nd most choice for RBC classification. A total of five different evaluation parameters have been used for WBC and RBC classification. Out of which accuracy, precision, and recall were primarily used with better results in percentage, as shown in Tables 7 and 9.

## Considerations of bias in dataset selection

The selection of data collection methods is a critical component of research that can introduce biases, which can affect the generalizability and applicability of the research. A prominent trend in the field of blood cell image analysis is the tendency towards manual data collection rather than the use of predefined datasets. This preference can be due to several factors that may introduce biases. Some of those factors are given below:

- **Specificity of research objectives:** Researchers may choose manual data collection to ensure that the dataset closely aligns with the specific objectives of their research. Predefined datasets may not sufficiently represent the particular features or conditions of interest.

**Dataset usage priority for WBC & RBC Classification**

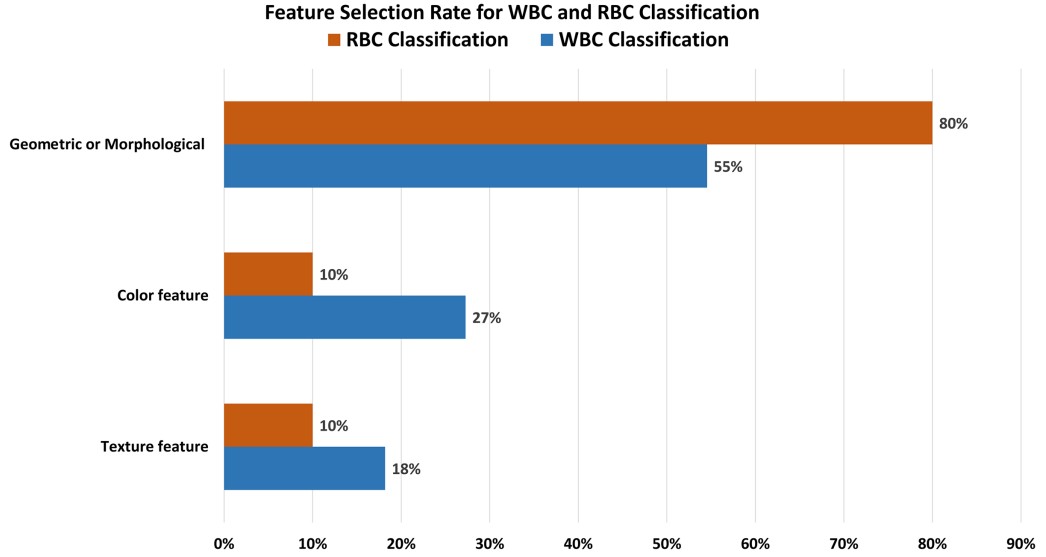

**Figure 9 Dataset usage rate for classification.** The comparative analysis of dataset selection for blood cell classification. Results show that most of the researchers use their own manually collected datasets instead of publicly available repositories.               

**Feature Selection Rate for WBC and RBC Classification**

**Figure 10 Feature selection for WBCs and RBCs.** The separate feature selection rate for WBC and RBC classification. The graph indicates that geometric or morphological features are most relevant for both RBC and WBC classification.               

- **Control over data quality:** Manual collection allows researchers to exercise greater control over the quality of data, including the resolution, contrast, and staining techniques, which are crucial for accurate image analysis.

- **Variability and diversity:** Predefined datasets often have limitations in terms of variability and diversity. Researchers choose the manual collection of data along with a broader range of variability in cell morphology. This phenomenon helped to develop an efficient CNN model for image processing.
- **Availability and accessibility:** The availability of large, annotated, and relevant datasets is still limited in the field of medical image analysis. Researchers may be compelled to collect data manually due to the lack of large-scale accessible datasets, which can result in a sampling bias.
- **Ethical and privacy concerns:** The use of predefined datasets may be restricted by ethical and privacy considerations, prompting researchers to collect data manually under controlled conditions that ensure compliance with ethical standards.

To reduce these biases, researchers must acknowledge and report the justification behind their choice of data collection methods. Additionally, employing a combination of predefined and manually collected datasets, when possible, could enhance the robustness of the experimental design. It is also important to consider cross-validation with independent datasets to confirm the efficiency and reliability of the findings.

## Comparison with existing work

The previously published surveys (*Das et al., 2022*; *Khan et al., 2021*; *SivaRao & Rao, 2023*; *Raina et al., 2023*; *Umamaheswari & Geetha, 2019*; *Asghar et al., 2023*) do not pay more attention to discussing the usability of the publicly available datasets. This issue is consistently highlighted as a significant challenge in developing and validating new models. Another common challenge mentioned in the surveys (*Das et al., 2022*; *Khan et al., 2021*; *Anilkumar, Manoj & Sagi, 2023*; *Raina et al., 2023*; *Rao & Rao, 2023*; *Umamaheswari & Geetha, 2019*; *More & Sugandhi, 2023*; *Al-Dulaimi & Makki, 2023*; *Veeraiah, Alotaibi & Subahi, 2023*) is the inherent complexity in microscopic images. This shows a demanding need for robust algorithms that can handle such complexities. Additionally, the inconsistency in diagnosis depending on the pathologist's experience is emphasized in surveys (*Das et al., 2022*; *Raina et al., 2023*; *Rao & Rao, 2023*; *Umamaheswari & Geetha, 2019*; *More & Sugandhi, 2023*; *Veeraiah, Alotaibi & Subahi, 2023*), highlighting the importance of automated systems for more consistent and accurate diagnosis. Each survey also brings unique contributions like (*SivaRao & Rao, 2023*) and (*Bhargavi et al., 2023*), which uniquely focus on using SegNet, EfficientNet, and XGBoost for WBC classification, providing a different perspective from other surveys. In *Bhatia et al. (2023)* authors pay attention to erythrocytes classification in sickle cell disease, pointing out gaps in the existing literature. The survey in *Veeraiah, Alotaibi & Subahi (2023)* focuses on a technical article that proposes a new method, the Histogram Threshold Segmentation Classifier (HTsC), rather than a traditional survey. Regarding weaknesses, some of the previous works have some common limitations, including (*Khan et al., 2021*; *Thomas & Sreejith, 2018*; *Byndur et al., 2023*; *Rao & Rao, 2023*; *Umamaheswari & Geetha, 2019*; *Al-Dulaimi & Makki, 2023*; *Asghar et al., 2023*), the absence of experimental results or comprehensive comparisons between methods. In contrast, the proposed survey

appears to be more comprehensive and holistic. It aims to cover both WBC and RBC images and addresses a wide range of challenges, including algorithmic efficiency, multi-modal Imaging, and explainability. Our work also explores areas often overlooked in existing literature, such as clinical validation, future directions, target audience, and real-world clinical trials. This comprehensive approach makes a valuable addition to the field, filling gaps in the existing literature and providing a more rounded view of the current challenges and future directions. The summery of the comparative analysis is shown in Table 10.

After a detailed survey on the blood cell segmentation and classification, the following research question may arise:

- **Algorithmic efficiency**: What are the most computationally efficient blood cell segmentation and Classification algorithms without sacrificing accuracy?
- **Multi-modal imaging**: How can different imaging modalities, like MRI, CT, and microscopy, be integrated for more robust blood cell analysis?
- **Explainability**: How can machine learning models be more interpretable to medical professionals not well-versed in computational techniques?
- **Real-time analysis**: Is it possible to develop real-time blood cell analysis systems that can be integrated into current medical workflows?
- **Algorithmic bias**: How can potential biases in machine learning algorithms be identified and mitigated, especially given the variations in blood cell morphology among different populations?
- **Automated feature selection**: Can automated feature selection techniques outperform human-selected features in the classification and segmentation of blood cells?
- **Telemedicine applications**: How can these technologies be adapted for remote diagnostics, particularly in resource-poor settings?
- **Clinical validation**: How do these machine learning models perform in clinical trials and what is their real-world applicability?

## ETHICAL CONSIDERATIONS IN AUTOMATED DIAGNOSIS AND PATIENT DATA PRIVACY

The following ethical considerations should be taken during the disease diagnosis procedure using machine learning and AI applications.

- **Patient consent and autonomy:** Informed consent is a key of ethical medical practice, ensuring that patients are aware of and agree to how their medical data is used in AI/ML applications. Autonomy is respected by allowing patients to make knowledgeable decisions about their participation in AI-driven diagnostics.
- **Data privacy and confidentiality:** Protecting patient data privacy involves implementing robust security measures to prevent data breaches and unauthorized access. Confidentiality is maintained by hiding patient data to prevent the exposure of personal health information from the general public.

**Table 10 Comparative analysis.** Summary of the previously published survey in the blood cell image segmentation and classification domain.

| Title | Focus area | Techniques analyzed | Challenges highlighted |
|---|---|---|---|
| *Das et al. (2022)* | WBC | SVM, KNN, ANN, CNN, RNN | Publicly accessible datasets, generalization, complexity |
| *Khan et al. (2021)* | WBC classification | Traditional machine learning (TML) and deep learning (DL) methods | Need for lightweight TML and DL techniques, transitioning from supervised to unsupervised learning |
| *SivaRao & Rao (2023)* | WBC classification | SegNet EfficientNet, and XGBoost | The conventional method is time-consuming, laborious, and potentially erroneous. |
| *Bhatia et al. (2023)* | RBC classification in sickle cell disease | Customized-DCNN, SHAP, and LIME for interpretability | Small unbalanced dataset, overlapping or clustered RBCs in some images |
| *Bhargavi et al. (2023)* | White blood cell classification | Deep-CNNs, decision tree | Traditional methods are time-consuming and less accurate |
| *Anilkumar, Manoj & Sagi (2023)* | Computer-aided diagnosis of leukemia | SVM, CNN, k-NN, naïve Bayes, ensemble classifiers | Lack of public datasets for chronic leukemia, Intra-observer and inter-observer variability |
| *Thomas & Sreejith (2018)* | White blood cells segmentation | K-means clustering, otsu thresholding, color-based segmentation | Complexity and uncertainty in microscopic blood smear images make WBCs segmentation challenging |
| *Byndur et al. (2023)* | Segmentation and classification WBCs | Staining techniques, datasets, preprocessing techniques, Otsu's method, K-means clustering, CNN, k-NN | Manual identification of WBCs is prone to errors, complexity in microscopic blood smear images |
| *Raina et al. (2023)* | Acute leukemia detection using deep learning | Preprocessing techniques like resizing, normalization, histogram equalization | Lack of publicly available datasets, complexity in microscopic blood smear images, variability in diagnosis depending on hematologist's experience. |
| *Rao & Rao (2023)* | WBC segmentation and classification | Pyramid scene parsing network MobilenetV3, artificial gravitational cuckoo search, ShufflenetV2 | Challenges in preprocessing include noise, occlusion, and missing data. |
| *Umamaheswari & Geetha (2019)* | Machine learning in leukemia detection | Otsu's method, automatic thresholding, Watershed Algorithm, k-means clustering | Lack of standard datasets for leukemia detection, Complexity in microscopic blood smear images |
| *More & Sugandhi (2023)* | Leukemia detection using machine learning | Noise removal, contrast adjustment, extraction methods for shape, texture, forest, naive bayes, SVM, and logistic regression. | Complexity in microscopic blood smear images, Challenges in preprocessing like noise, occlusion, missing data. |
| *Al-Dulaimi & Makki (2023)* | Blood cell detection and classification in CAD systems | Preprocessing techniques like noise removal, contrast adjustment. Segmentation methods like Fuzzy C means, K-means clustering, and thresholding. | Different staining techniques affecting segmentation, Non-uniform illumination, Variation in cell maturity stages, And morphology complexities. |
| *Asghar et al. (2023)* | Medical image analysis for white blood cell classification | Preprocessing techniques, feature extraction, CNN, R-CNN, Fast R-CNN, GAN. | Availability of appropriate datasets, medical training of researchers for better understanding of WBC structure |
| *Veeraiah, Alotaibi & Subahi (2023)* | Medical image analysis for leukemia detection | Histogram threshold segmentation classifier (HTsC) | Overburdening of pathologists with large data sets, variations in illumination and staining in manual setups |
| Proposed Survey | Target both WBC and RBC images | Machine learning and deep learning | Multidimensional research question and future directions, target audience |

- **Bias and fairness:** Bias in AI/ML can lead to unsatisfactory treatment outcomes. Ensuring fairness involves the careful curation and annotation of datasets and algorithm design to prevent discriminatory practices and promote reasonable healthcare services.

- **Accountability and transparency:** Accountability in AI/ML systems means having clear responsibilities for CNN model outcomes, including errors. Transparency is about making the AI decision-making process understandable to users and patients, ensuring trust in automated diagnostics. Transparency can be achieved with the integration of explainable AI with current AI models.

- **Impact on clinical practice:** The integration of AI and ML technologies into healthcare settings has the potential to transform the traditional dynamics between clinicians and patients. It's essential to consider how these technologies will affect clinical workflows and patient interactions.

- **Patient data governance:** Data governance refers to the policies and processes that ensure the ethical collection, management, and use of patient data. It is very important to establish a framework that maintains the integrity and privacy of patient information within AI/ML applications.

## CONCLUSION AND FUTURE DIRECTION

This work started with a question that could be debatable. However, different laboratory tests are currently used for diagnosing blood-related diseases, and morphological analysis of blood elements has a key contribution to the diagnostic process. This work explores state-of-the-art techniques and models developed for blood image analysis. We have also performed several comparative studies to determine the best techniques for blood cell segmentation, classification, and feature extraction. Analysis reveals that 50% of previous works used manually collected blood cell images to segment WBCs and RBCs. To achieve precise ROI regarding WBC segmentation, 80% of researchers selected morphological (40%) and texture (40%) features. While in the case of RBC, the researcher prioritized geometric features. Comparative analysis on feature extraction discloses that the researcher gave precedence to morphological features (55%) for segmentation and classification. For the classification of WBCs, 45% of the previous work used the ALL-IDB dataset, while 36% of the literature relied on manually collected images. However, in RBC classification, 73% of researchers rely on manual image collection. Morphological features got higher priority, *i.e.*, 80% and 55% for RBC and WBC classification, respectively, compared to texture and color features. It is concluded that researchers can get efficient performance regarding the segmentation of WBCs and RBCs with specified image collection and texture or morphological feature selection. While for WBC and RBC classification, better performance can be attained using the ALL-IDB dataset along with morphological feature selection and manual image collection with morphological feature selection, respectively. The diagnostic accuracy for blood-related diseases like leukemia, anemia, lymphoma, and thalassemia can be significantly enhanced through the effective use of CAD techniques, which have evolved considerably in recent years. This survey provides a broad and in-depth review of the techniques being employed, from image segmentation to classification,

feature selection, utilization of evaluation matrices and dataset selection. The inconsistency in dataset selection suggests a need for standardized, high-quality datasets to further strengthen these techniques' diagnostic capabilities. Additionally, the popularity of morphological features indicates that future research could further explore and innovate in this direction. On the above discussion, following future work direction should be recommended.

- **Benchmarking studies**: Future research should focus on creating benchmark studies that can objectively compare the performance of various algorithms under standardized conditions.
- **Collaborative datasets**: There is a need for larger, more diverse, and publicly available datasets that can be used to train more robust and generalizable models.
- **Interdisciplinary collaboration**: Building teams of researchers that include hematologists and data scientists that can provide more holistic solutions.
- **Ethical guidelines**: As machine learning algorithms become more complex, creating ethical guidelines specifically for blood cell segmentation and Classification will be increasingly important.
- **Localized models**: Research into developing models that are specifically trained on data from certain demographics or geographical locations can improve the applicability and accuracy of these systems.
- **Human-in-the-loop systems**: Designing systems where machine learning algorithms work with human experts could be a fruitful avenue, balancing the strengths and weaknesses of both.
- **Real-world clinical trials**: Moving beyond laboratory settings to test these algorithms in real-world clinical scenarios will be crucial for widespread adoption.

### Funding
This material is based upon work supported by the Higher Education Commission (HEC) Pakistan through the National Research Program for Universities (NRPU) under Project ID 16017. The funders had no role in study design, data collection and analysis, decision to publish, or preparation of the manuscript.

### Grant Disclosures
The following grant information was disclosed by the authors:
Higher Education Commission (HEC).
National Research Program for Universities (NRPU): 16017.

### Competing Interests
The authors declare that they have no competing interests.

## Author Contributions

- Muhammad Shahzad conceived and designed the experiments, performed the experiments, performed the computation work, prepared figures and/or tables, authored or reviewed drafts of the article, and approved the final draft.
- Farman Ali conceived and designed the experiments, analyzed the data, authored or reviewed drafts of the article, and approved the final draft.
- Syed Hamad Shirazi conceived and designed the experiments, performed the experiments, performed the computation work, prepared figures and/or tables, and approved the final draft.
- Assad Rasheed conceived and designed the experiments, performed the experiments, performed the computation work, prepared figures and/or tables, and approved the final draft.
- Awais Ahmad performed the experiments, performed the computation work, prepared figures and/or tables, authored or reviewed drafts of the article, and approved the final draft.
- Babar Shah analyzed the data, authored or reviewed drafts of the article, and approved the final draft.
- Daehan Kwak conceived and designed the experiments, analyzed the data, authored or reviewed drafts of the article, and approved the final draft.

## Data Availability

This is a literature review.

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
