# Peer review of "Blood cell image segmentation and classification: a systematic review"

_PeerJ Computer Science, doi:10.7717/peerj-cs.1813_

## Round 0.1 · original submission · Major Revisions

The authors should revise according to the reviewers' comments.

**Language Note:** The review process has identified that the English language must be improved. PeerJ can provide language editing services - please contact us at copyediting@peerj.com for pricing (be sure to provide your manuscript number and title). Alternatively, you should make your own arrangements to improve the language quality and provide details in your response letter. – PeerJ Staff

Reviewer 1 ·

Basic reporting

The paper titled "Blood Cell Image Segmentation and Classification: A Systematic Review" aims to provide a comprehensive review of existing literature on blood cell image analysis using deep learning techniques, with a focus on segmentation and classification of white blood cells (WBC) and red blood cells (RBC). Overall, the provided sections of the paper demonstrate a well-structured and informative approach to the survey. The authors effectively communicate the importance of the topic, the motivation for their research, and the intended audience for their findings. The abstract provides a concise summary of the key aspects of the survey, while the introduction and motivation sections provide a more detailed background and rationale for the study. But before final submission, authors are required to consider the following comments:
Introduction and Research Questions: Your paper starts with a research question, which is a good approach. However, you might consider adding a more detailed introduction to provide context and motivation for your research questions. Explain why blood cell segmentation and classification are important and briefly discuss the challenges in this field.

Experimental design

Ethical Considerations: Given the increasing use of machine learning and AI in medical applications, it might be worthwhile to include a brief section on ethical considerations. Discuss potential ethical challenges related to automated diagnosis and patient data privacy.

References: Make sure to review and ensure the consistency of your references. Follow a specific citation style (e.g., APA, IEEE, etc.) consistently throughout the paper.

Proofreading: Carefully proofread the paper for any typographical or grammatical errors. A well-edited paper enhances its professionalism and readability.

Validity of the findings

Datasets and Benchmarks: Provide more details about the datasets and benchmarks used for testing these methods. Mention the size of the datasets, their sources, and any preprocessing steps applied. This will help readers understand the context in which these methods were evaluated.

Cite this review as

Reviewer 2 ·

Basic reporting

This survey paper offers a comprehensive overview of the current research on blood image analysis using deep learning techniques. It addresses various aspects of the field, from segmentation to classification, and highlights the importance of considering multiple components in the diagnostic process. The review is within the scoope of the journal.

The language is clear but it requires a thorough proofread. The searched liturature provides an ample backgroud of the field.

Experimental design

It is required to discuss potential biases associated with researchers' choices in the methods section (add a new subsection). For example, why do many researchers choose manual data collection over predefined datasets?

Validity of the findings

Discussion on feature extraction of blood elements is vague. Elaborate more on this section, focusing on feature exraction of WBCs and RBCs. So that the reader can understand the importance of features selection separately of WBCs and RBCs

Tables 2, 4, 5, 6, 7 and Figures 3, 6, 7 require a detailed caption. Provide detalied description of all figures and tables.

Additional comments

None

Cite this review as

---

## Round 0.2 · accepted · Accept

The authors have provided satisfactory comments to the reviewers' concerns. The article is accepted based on revisions.

Reviewer 1 ·

Basic reporting

Good

Experimental design

Good

Validity of the findings

Good

Additional comments

The authors have addressed all my concerns.

Cite this review as

Reviewer 2 ·

Basic reporting

The review fits well within the journal's scope. The focus is on blood cell image segmentation and classification, which has broad cross-disciplinary applications. The introduction, motivation, and target audience are clear from the manuscript.

Experimental design

A good coherent read with connectivity between paragraphs and subsections. The methodology is consistent and comprehensive, and the sources are cited correctly where required.

Validity of the findings

Yes.
Yes.

Additional comments

None

Cite this review as